# A global gridded (0.1º x 0.1º) inventory of methane emissions from oil, gas, and coal exploitation based on national reports to the United Nations Framework Convention on Climate Change

Tia R. Scarpelli[1], Daniel J. Jacob[1], Joannes D. Maasakkers[1], Melissa P. Sulprizio[1], Jian-Xiong Sheng[1],
Kelly Rose[2], Lucy Romeo[2], John R. Worden[3], and Greet Janssens-Maenhout[4]

[1]Harvard University, Cambridge, MA 02138, United States
[2]U.S. Department of Energy, National Energy Technology Laboratory, Albany, OR 97321, United States
[3]Jet Propulsion Laboratory, California Institute of Technology, Pasadena, CA 91109, United States
[4]European Commission Joint Research Centre, Ispra (Va), Italy

*Correspondence to*: Tia R. Scarpelli (tscarpelli@g.harvard.edu)

**Abstract.** Individual countries report national emissions of methane, a potent greenhouse gas, in accordance with the United Nations Framework Convention on Climate Change (UNFCCC). We present a global inventory of methane emissions from oil, gas, and coal exploitation that spatially allocates the national emissions reported to the UNFCCC (Scarpelli et al., 2019). Our inventory is at 0.1° x 0.1° resolution and resolves the subsectors of oil and gas exploitation, from upstream to downstream, and the different emission processes (leakage, venting, flaring). Global emissions for 2016 are 41.5 Tg a$^{-1}$ for oil, 24.4 Tg a$^{-1}$ for gas, and 31.3 Tg a$^{-1}$ for coal. An array of databases is used to spatially allocate national emissions to infrastructure including wells, pipelines, oil refineries, gas processing plants, gas compressor stations, gas storage facilities, and coal mines. Gridded error estimates are provided in normal and lognormal forms based on emission factor uncertainties from the IPCC. Our inventory shows large differences with the EDGAR v4.3.2 global gridded inventory both at the national scale and in finer-scale spatial allocation. It shows good agreement with the gridded version of the United Kingdom's National Atmospheric Emissions Inventory (NAEI). There are significant errors on the 0.1° x 0.1° grid associated with the location/magnitude of large point sources but these are smoothed out when averaging the inventory over a coarser grid. Use of our inventory as prior estimate in inverse analyses of atmospheric methane observations allows investigation of individual subsector contributions and can serve policy needs by evaluating the national emissions totals reported to the UNFCCC. Gridded data sets can be accessed at https://doi.org/10.7910/DVN/HH4EUM.

## 1 Introduction

Methane is the second most important anthropogenic greenhouse gas after $CO_2$, with an emission-based radiative forcing of 1.0 W m$^{-2}$ since pre-industrial times as compared to 1.7 W m$^{-2}$ for $CO_2$ (Myhre, 2013). Major anthropogenic sources of methane include the oil/gas industry, coal mining, livestock, rice cultivation, landfills, and wastewater treatment. Individual countries must estimate and report their anthropogenic methane emissions by source to the United Nations in accordance

with the United Nations Framework Convention on Climate Change (UNFCCC, 1992). These estimates rely on emission factors (amount emitted per unit of activity) that can vary considerably between countries in particular for oil and gas (Larsen, 2015). This variation may reflect differences in infrastructure between countries but also large uncertainties (Allen et al., 2015; Brantley et al., 2014; Mitchell et al., 2015; Omara et al., 2016; Robertson et al., 2017), including a possible under-accounting of abnormally high emitters (Duren et al., 2019; Zavala-Araiza et al., 2015).

Top-down inverse analyses of atmospheric methane observations can provide a check on the national emission inventories (Jacob et al., 2016), but they require prior information on the spatial distribution of emissions within the country. This information is not available from the UNFCCC reports. The EDGAR global emission inventory with 0.1° x 0.1° gridded resolution (European Commission, 2011, 2017) has been used extensively as prior estimate for methane emissions in inverse analyses. EDGAR prioritizes the use of a consistent methodology between countries for emissions estimates, including the use of IPCC Tier 1 methods (IPCC, 2006), and then spatially distributes emissions using proxy data like satellite observations of gas flaring (Janssens-Maenhout et al., 2019). However, its oil/gas emissions show large differences compared to inventories that utilize more detailed data specific to a country or region (Jeong et al., 2014; Lyon et al., 2015; Maasakkers et al., 2016; Sheng et al., 2017). The latest public version, EDGAR v5.0 (Crippa et al., 2019; European Commission, 2019), provides separate gridded products for oil, gas, and coal exploitation emissions for each year from 1970 to 2015 but with no further subsector breakdown. Some other regional and global multi-species emission inventories also include methane but have coarse spatial and/or sectoral resolution, such as CEDS (Hoesly et al., 2018), REAS (Kurokawa et al., 2013), or GAINS (Höglund-Isaksson, 2012). Gridded emission inventories for the oil, gas, and coal sectors with subsectoral and/or point source information have been produced for individual production fields (Lyon et al., 2015), California (Jeong et al., 2014; Jeong et al., 2012; Zhao et al., 2009), and a few countries including Australia (Wang and Bentley, 2002), Switzerland (Hiller et al., 2014), the United Kingdom (Defra and BEIS, 2019), China (Peng et al., 2016; Sheng et al., 2019), the US (Maasakkers et al., 2016), and Canada and Mexico (Sheng et al., 2017).

Here we create a global 0.1° x 0.1° gridded inventory of methane emissions from the oil, gas, and coal sectors, resolving individual activities (subsectors) and matching national emissions to those reported to the UNFCCC. The inventory effectively provides a spatially downscaled representation of the UNFCCC national reports by attributing the national emissions to the locations of corresponding infrastructure. Our premise is that the national totals reported by individual countries contain country-specific information that may not be publicly available or easily accessible. In addition, the UNFCCC national reports provide the most policy-relevant estimates of emissions to be evaluated with results from top-down inverse analyses. Our downscaling relies on global data sets for oil/gas infrastructure locations available from Enverus (2017), Rose (2017), the National Energy and Technology Laboratory's Global Oil & Gas Infrastructure (GOGI) inventory and geodatabase (Rose et al., 2018; Sabbatino et al., 2017), and other sources. National emissions from coal mining are

distributed according to mine locations from EDGAR v4.3.2. We present results for 2016 which is the most recent year available from the UNFCCC, but our method is readily adaptable to other years.

## 2 Data and methods

### 2.1 National emissions data

Figure 1 gives a flow chart of the emission processes from oil, gas, and coal exploitation as resolved in our inventory. The emissions characterized here correspond to the IPCC (2006) category "fugitive emissions from fuels" (category code 1B). Here and elsewhere we refer to "sectors" as oil, gas, or coal. We refer to "subsectors" as the separate activities for each sector resolved in Fig. 1, e.g., "Gas production". The subsectors were chosen to match UNFCCC reporting as much as possible. We refer to "processes" as the means of emission which can be leakage, venting, or flaring. Leakage emissions

include all unintended emissions such as from equipment leaks, evaporation losses, and accidental releases. Coal emissions are lumped together, including contributions from surface and underground mines during mining and post-mining activities (IPCC, 2006), without further partitioning because the emissions are mainly at the locations of the mines. We create a separate gridded inventory file for each sector, subsector, and process as specified by the individual boxes of Fig. 1. The subsectors reported by countries to the UNFCCC vary, so our first step is to compile national emissions for each subsector

and process listed in Fig. 1 so that emissions can then be allocated spatially as described in Sect. 2.2.

### 2.1.1 UNFCCC reporting

The UNFCCC receives inventory reports from 43 developed countries as 'Annex I' parties and communications from 151 countries as 'non-Annex I' parties. The 43 Annex I countries report annually and disaggregate emissions to subsectors. Non-Annex I countries report total emissions for the combined oil/gas sector and total emissions for the coal sector, but they are

not required to report annually or to disaggregate emissions by subsectors. We use the UNFCCC GHG Data Interface as of May 2019 (UNFCCC, 2019) to download emissions reported by Annex I countries for the year 2016 and emissions reported by non-Annex I countries for the year 2016 if available or the most recent year if not.

Annex I countries report oil/gas leakage emissions by subsector, and these emissions can be used in the inventory as

reported. An exception is for gas transmission and gas storage which are only reported as a combined total and have to be disaggregated. Also, Annex I venting and flaring emissions are only reported as sector totals (oil venting, oil flaring, gas venting, and gas flaring) which have to be disaggregated to the subsectors of Fig. 1. Annex I countries may choose to report emissions for a given subsector as "Included Elsewhere" which means the emissions have been included in the emissions total reported for a different subsector. The most common example is when venting and flaring emissions are included

within reported leakage emissions as is the case for oil/gas venting and flaring in the United States. We do not attempt to separate these emissions here because it does not affect the spatial allocation of emissions.

The emissions reported for oil/gas and coal by non-Annex I countries have to be disaggregated to the subsectors of Fig. 1. If a non-Annex I country does not report coal emissions separate from oil/gas we treat it as a non-reporting country (Sect. 2.1.3). Some non-Annex I countries choose to report oil/gas emissions by subsectors similar to Annex I countries. These reported emissions are not available in the GHG Data Interface and require inspection of reports submitted by each country, including National Communications (submitted every 4 years; COP, 2002) and Biennial Update Reports (submitted every 2 years; COP, 2011). We inspect reports for countries with estimated or reported oil+gas emissions greater than or equal to 1 Tg a$^{-1}$. These countries are Algeria, Brazil, China, India, Indonesia, Iran, Iraq, Malaysia, Nigeria, Qatar, Saudi Arabia, Uzbekistan, and Venezuela. The extent of emissions disaggregation by subsector for these non-Annex I countries varies. Algeria, India, Malaysia, Nigeria, Saudi Arabia, and Uzbekistan report similarly to Annex I countries while other countries only provide limited disaggregation.

### 2.1.2 Disaggregation by subsectors and processes

We disaggregate reported emissions as needed by estimating emissions for each subsector and process using IPCC Tier 1 methods (IPCC, 2006) and then applying these relative subsector/process contributions to the reported emissions. We multiply the IPCC Tier 1 emission factor for each subsector to national activity data from the U.S. Energy and Information Administration (EIA, 2018a). Oil production volume is used as activity data for emissions from oil exploration and production, while volume of oil refined is used for oil refining emissions. Oil transported by pipeline (oil production + imported volume) is used for oil transport leakage emissions and 50% of oil production volume is used for oil transport venting emissions (assumed to occur during truck and rail transport). Total gas production volume is used as activity data for gas production and processing; marketable gas volume (consumed gas + exported gas) is used for gas transmission and storage; and gas consumption is used for gas distribution. Disaggregated subsector emissions from non-Annex I countries are then adjusted to 2016, if necessary, using the EIA activity data.

We disaggregate Annex I venting and flaring emissions using the relative contribution of each subsector to total venting or flaring as estimated by IPCC Tier 1 methods. We cannot do this for the exploration or oil refining subsectors because IPCC methods do not separate venting and flaring emissions from leaks. Instead we compare the IPCC estimate for total emissions from each subsector (leakage + venting + flaring) with the reported leakage emissions. If the IPCC emissions total is greater than the reported leakage emissions we assume that the excess emissions can be attributed to venting and flaring. Venting and flaring emissions from gas storage and gas distribution similarly cannot be separated from leaks, but we assume that leaks dominate these subsectors.

### 2.1.3 Non-reporting countries

For the few countries that do not report emissions from oil, gas, and coal to the UNFCCC we estimate emissions following IPCC Tier 1 methods applied to the 2016 EIA activity data. This is the case notably for Libya and Equatorial Guinea which both have total emissions greater than 0.1 Tg a$^{-1}$. We also use this method for countries that do not separately report coal and oil/gas emissions, notably Angola which is the only such country that has total emissions greater than 0.1 Tg a$^{-1}$. For the countries that do not have EIA activity data, notably Uganda and Madagascar which account for most of the pipelines and wells in such countries, we use the infrastructure data described in Sect. 2.2 together with the average emissions per infrastructure element based on countries that do report emissions.

### 2.1.4 Coal emissions

For coal, Annex I emissions for 2016 are used as reported. Non-Annex I emissions reported as total coal emissions are adjusted to 2016 as needed using activity data provided by the EIA (2018a). For the few countries that do not report to the UNFCCC, we use the coal emissions data embedded in EDGAR v4.3.2 Fuel Exploitation with additional information from EDGAR to separate coal from oil/gas; these countries account for less than 1% of global coal emissions.

### 2.2 Spatially mapping emissions

Our next step is to allocate the national emissions from each subsector of Fig. 1 spatially on a 0.1° x 0.1° grid. National emissions are allocated following the procedure described below for all countries. An exception is for the contiguous US (Maasakkers et al., 2016) and for oil/gas in Canada and Mexico (Sheng et al., 2017), where we use existing inventories constructed for 2012 on the same 0.1° x 0.1° grid and scaled here by subsector to match the corresponding national UNFCCC reports for 2016. The two North American inventories only provide a total oil emissions gridded product, and we simply scale this product to match the reported subsector totals for oil emissions. Alaska is missing from the US inventory so we estimate emissions for Alaska using the EPA State Inventory Tool (EPA, 2018) following the methods outlined in the Alaska Greenhouse Gas Emission Inventory (Alaska Department of Environmental Conservation, 2018) and apply the procedures described below to distribute these emissions spatially. Other previously reported gridded national emission inventories are not used here due to their limited spatial resolution and/or limited disaggregation of emissions (Hiller et al., 2014; Höglund-Isaksson, 2012; Kurokawa et al., 2013; Wang and Bentley, 2002). We will use the gridded version of the UK National Atmospheric Emissions Inventory (NAEI; Defra and BEIS, 2019) as independent evaluation of our inventory in Sect. 3.3.

### 2.2.1 Allocating upstream emissions to wells

Upstream emissions, including exploration and production, are allocated spatially to wells as illustrated in Fig. 2. Our principal source information on wells is Enverus (2017). It provides worldwide point locations of onshore and offshore wells, well activity status, and well content. Well activity status is used to separate active from inactive wells. Inactive wells

are assumed not to emit. Well content is used to separate oil and gas wells, though this separation can be difficult as oil wells also have production of associated gas. We label wells as unknown content if their content is either unavailable or not clearly defined as oil or gas (this makes up approximately 24% of Enverus wells outside North America). We uniformly distribute emissions over the appropriate wells in each country. Within each country we determine the percentage of wells with unknown content and uniformly distribute this percentage of total oil and gas upstream emissions to those wells. We then uniformly distribute the remaining oil and gas emissions to oil and gas wells, respectively.

Well data are missing from Enverus for a number of countries. An alternative global well database with wells drilled up to 2016 is available from Rose (2017) based on a combination of open source data and proprietary data from IHS Markit (2017) and mapped on a 0.1º x 0.1º grid (total number of wells per grid cell). The Rose database does not include information on oil versus gas content. We use this database for all countries that are either missing from the Enverus database or for which the Rose database has 50% greater number of wells than Enverus. This includes 47 of the 134 countries with active well infrastructure. Of those 47 countries, the ones with the greatest number of wells are Russia, United Arab Emirates (UAE), China, Libya, Saudi Arabia, Turkmenistan, Ukraine, and Azerbaijan. We distribute total upstream emissions from both oil and gas uniformly over all active wells within each country. The Rose database includes offshore wells, but they are not identified by country so we rely solely on Enverus for offshore wells. Between the Enverus and Rose databases, over 99% of global upstream emissions can be spatially allocated. The rest are allocated along pipelines.

### 2.2.2 Allocating emissions to midstream infrastructure

Midstream emissions from oil refining, oil transport, gas processing, gas transmission, and gas storage within a given country are allocated using GOGI infrastructure locations (Rose et al., 2018; Sabbatino et al., 2017) as shown in Fig. 2. Spatial information on non-well infrastructure in Alaska is taken from the U.S. Energy Mapping System (EIA, 2018b). Oil refining emissions are attributed evenly to refinery locations within a given country. Oil transport emissions can occur during pipeline, truck, or tanker transport but we assume that they are mainly along pipelines and allocate them by pipeline length on a 0.1º x 0.1º grid. Gas processing, transmission, and storage emissions are distributed uniformly among the processing plant, compressor station, and storage facilities, respectively, in each country. Annex I countries report "Other" emissions for oil and gas which are distributed equally to wells and pipelines.

The GOGI database was created through a machine-learning web search of public databases for mention of oil and gas infrastructure, so it is limited to open-source information available as of 2017. It misses some infrastructure locations (Rose et al., 2018), so the spatial allocation of emissions within a country may be biased to the identified locations. To alleviate this problem, we check each country for exceedance of an oil or gas volume-per-facility threshold (e.g., volume of gas processed per day per processing plant). These thresholds, given below, are conservative in that they are based on the world's largest facilities or the upper limit of infrastructure design. If the threshold is exceeded we estimate the percentage of facilities

missing, and the corresponding percentage of subsector emissions is allocated to pipelines since non-well infrastructure tends to lie along pipeline routes. Visual inspection suggests that countries with pipelines in the GOGI database are not missing any significant pipeline locations which is consistent with a gap analysis for that database (Rose et al., 2018).

For oil refining in each country, we determine a refining rate per refinery by distributing the total volume of oil refined (EIA, 2018a) for 2016 over the GOGI refineries in that country. If the refining rate exceeds the threshold set by the Jamnagar Refinery in India of 1.24 million barrels of crude oil per day (Duddu, 2013), then oil refining emissions corresponding to the missing refineries are allocated to pipelines. The same is done for processing plants, storage facilities, and compressor stations. Processing plants are missing if production of natural gas (EIA, 2018a) distributed over processing plants exceeds

57 million cubic meters of gas per day per facility based on the Ras Laffan processing plant in Qatar (Hydrocarbons-Technology, 2017). Storage facilities are missing if production of marketable gas (EIA, 2018a) exceeds 68 billion cubic feet per year per facility, corresponding to the total US capacity determined from marketable gas volume and number of active storage facilities (EIA, 2015). Compressor stations are missing if the implied gas pipeline length (CIA, 2018) between GOGI stations is more than 100 miles.

We separate the GOGI pipelines into oil and gas when possible though a significant number have unknown content. For each country, we determine the percentage of pipelines with unknown content and distribute this percentage of total oil and gas pipeline emissions to those pipelines. The remaining oil and gas emissions are allocated to oil and gas pipelines, respectively. In order to avoid allocating Russia's significant gas transmission emissions to unknown content pipelines, we

instead use a gridded 0.1º x 0.1º map of gas pipelines based on the detailed Oil & Gas Map of Russia/Eurasia & Pacific Markets (Petroleum Economist Ltd, 2010).

### 2.2.3 Allocating downstream emissions

Downstream gas distribution emissions are associated with residential and industrial gas use. We allocate these emissions within each country on the basis of population using the Gridded Population of the World (GPW) v4.10 30 arc second map

(CIESIN, 2017) for 2010. Midstream emissions are also allocated to population for countries missing in the GOGI database (<1% of global midstream emissions).

### 2.2.4 Allocating coal emissions

Coal mining and post-mining emissions from individual countries are allocated spatially to mines based on EDGAR v4.3.2 emission grid maps for 2012 (0.1° x 0.1° resolution). A specific inventory for China shows a greater number of mines than

EDGAR v4.3.2 (Sheng et al., 2019), but to the authors' knowledge EDGAR is the only fine resolution database of coal mine locations with global coverage. EDGAR v4.3.2 estimates surface and underground mine emissions separately but distributes

them to mines as a combined total, so emissions from both types of mines are combined here. Alaskan emissions are allocated to Alaska's single operational coal mine, Usibelli (EIA, 2018b).

## 2.3 Error estimates

Inverse analyses of atmospheric methane observations require error estimates on the prior emission inventories as a basis for
Bayesian optimization (Jacob et al., 2016). Here we use uncertainty ranges from IPCC (2006) to estimate error standard deviations in our inventory. The IPCC reports relative uncertainty ranges for the emission factors used in Tier 1 national estimates, as summarized in Table 1. Uncertainties in national estimates are dominated by emission factors (typically 50-100%) as compared to the better known activity data (5-25%; IPCC, 2006). The emission factor uncertainties in Table 1 correspond to the subsectors and processes of our inventory (Fig. 1). We differentiate between Annex I and non-Annex I
countries based on IPCC uncertainty ranges for 'Developed' and 'Developing' countries.

In the absence of better information, we interpret the IPCC uncertainty ranges as representing the 95% confidence intervals. The ranges are generally asymmetric, but we approximate them in Table 1 in terms of either (1) a relative error standard deviation (RSD) assuming a normal error probability density function (pdf), or (2) a geometric error standard deviation
(GSD) assuming a lognormal error pdf. The 95% confidence interval then represents a range of 4 standard deviations (± 2 standard deviations) in linear space for the normal pdf and in log space for the lognormal pdf. For the assumption of a normal error pdf, we take the average of the IPCC upper and lower uncertainty limits for each subsector and halve this value to get the RSD with an allowable maximum RSD of 100%. For the assumption of a lognormal error pdf, the IPCC limits are log-transformed, halved, averaged and transformed back to linear space to yield the GSD for the lognormal error pdf. The
lower limits of the IPCC uncertainty ranges are capped at 90% when determining the GSD. We provide both normal and log-normal error standard deviations in our inventory, as both may be useful for inverse analyses. Assuming a normal error pdf has the advantage of providing a proper model of mean emissions, while assuming a log-normal error pdf has the advantage of enforcing positivity and better allowing for anomalous emitters (Maasakkers et al., 2019).

We assume that our large relative errors at the national scale can be applied directly to the 0.1° x 0.1° grid for lack of better information. An error analysis for the gridded EPA inventory based on comparison to a more detailed inventory for Northeast Texas (Barnett Shale) showed that the relative error in emissions from oil systems was not significantly higher on the 0.1° x 0.1° grid than the national error estimate of 87%, while the relative error for gas systems on the 0.1° x 0.1° grid was twice the national estimate of 25% (Maasakkers et al., 2016). That work further showed that displacement error due to
spatial misallocation of emissions is negligibly small as long as the error in emission location is considered isotropic. Further error evaluation is presented in Section 3.3 by comparison to the UK's independently developed gridded national inventory.

# 3 Results & discussion

## 3.1 Global, national, and grid scale emissions

Table 2 lists the 2016 global methane emissions from oil, gas, and coal, broken down by the subsectors and processes resolved in our inventory. Total emission from fuel exploitation is 97.2 Tg a$^{-1}$ including 41.5 Tg a$^{-1}$ from oil, 24.4 Tg a$^{-1}$ from gas, and 31.3 Tg a$^{-1}$ from coal. Oil emissions are mainly from production, in part because oil fields often lack the capability to capture associated gas. Gas emissions are distributed over the upstream, midstream, and downstream subsectors. EDGAR v4.3.2 has a similar global total for fuel exploitation (107 Tg a$^{-1}$), but the spatial distribution is very different as shown in Sect. 3.2. Top-down inverse analyses compiled by the Global Carbon Project give a range of 90-137 Tg a$^{-1}$ for fuel exploitation in 2012 (Saunois et al., 2016).

Figure 3 lists the top 20 emitting countries for oil, gas, and coal in our inventory. These account for over 90% of each sector's global emission. The largest emissions are from Russia for oil, the US for gas, and China for coal. Oil and gas emissions for individual countries tend to be dominated by one of the two fuels. Notable exceptions are Russia, the US, Iran, Canada, and Turkmenistan which have large contributions from both. Annex I and non-Annex I countries reporting to the UNFCCC account for 49% and 47% of global emissions, respectively, with the remaining 4% of emissions contributed by countries that do not report to the UNFCCC.

Figure 4 shows the global distribution of methane emissions separately for oil, gas, and coal. Oil emissions are mainly in production fields. Contributions from gas production, transmission, and distribution can all be important with the dominant subsector varying between countries. The highest emissions are from oil/gas production fields, gas transmission routes, and coal mines. Most emissions along gas transmission routes are from compressor stations, processing plants, and storage facilities.

## 3.2 Comparison to the EDGAR v4.3.2 global gridded inventory

Figure 5 compares the spatial distribution of our 2016 emissions from fuel exploitation (sum of the oil, gas, and coal emissions from Fig. 4) to the corresponding 2012 emissions in EDGAR v4.3.2 (European Commission, 2017; Janssens-Maenhout et al., 2019). There are large differences between the inventories in terms of spatial patterns within each country, due to differences in both subsector contributions and spatial allocation of these contributions. Emissions along pipelines are generally lower in our work and emissions from production fields are generally higher. EDGAR v4.3.2 has more of a tendency to allocate midstream emissions to pipelines rather than to specific facilities.

National total emissions in our inventory (based on UNFCCC reports) are also very different from EDGAR. Figure 6 compares national emissions from fuel exploitation as reported to the UNFCCC versus EDGAR v4.3.2 emissions in the

same year. Russia, Venezuela, and Uzbekistan report emissions that are more than a factor of 2 greater than EDGAR v4.3.2. Iraq, Qatar, and Kuwait report emissions that are more than an order of magnitude lower than EDGAR v4.3.2 though their last reporting years are old (1997, 2007, and 1994, respectively). The discrepancies between our work and EDGAR v4.3.2 in Russia and the Middle East lead to a greater emissions contribution from high latitudes and a lesser contribution from low latitudes in the Northern Hemisphere in our work.

The causes of differences between the UNFCCC national totals used in our work and EDGAR v4.3.2 are country and subsector specific because each country may choose to use a methodology for emissions estimation that differs from default methods. Emission factors per unit of activity inferred from the UNFCCC reports can vary by orders of magnitude between countries (Larsen, 2015). This may reflect real differences in regulation of venting and flaring (especially for oil production), maintenance and age of infrastructure, and the size and number of facilities within a country. For example, Middle East countries report low emissions relative to their production volumes and this may reflect a tendency to have a small number of high-producing wells. In contrast, Russia and Uzbekistan report high emissions relative to oil production and gas processing volumes. For Russia, differences may be due to the inclusion of accidental releases in UNFCCC reporting which are not considered by EDGAR v4.3.2 (Janssens-Maenhout et al., 2019). Russia also reports large emissions from intentional venting. The National Report of Uzbekistan (2016) attributes their high emissions to leaky infrastructure and recent increases in produced and transported gas volumes which may lead to operation of equipment at over-capacity. Beyond these considerations, there may also be large errors in the emission estimates reported by individual countries to the UNFCCC. Inverse analyses of atmospheric methane observations using our inventory as prior estimate would provide insight into these errors.

### 3.3 Comparison to the United Kingdom national gridded inventory

The UK Department for Environment, Food, and Rural Affairs (Defra) and Department for Business, Energy, and Industrial Strategy (BEIS) produce a gridded version of their annual National Atmospheric Emissions Inventory (NAEI) with 0.01° x 0.01° resolution. This provides an opportunity for evaluating our spatial allocation of emissions since the allocation in the NAEI inventory is better informed by local data, including direct reporting of emissions by large emitters which account for 35% of fuel exploitation emissions.

Figure 7 compares our inventory for fuel exploitation to the most recent version of the NAEI in 2017 (Defra and BEIS, 2019) and to EDGAR v4.3.2 in 2012 (European Commission, 2017). National totals are identical in our inventory and the NAEI, as would be expected since the NAEI is used for UNFCCC reporting. The EDGAR v4.3.2 national total agrees with the 2012 NAEI (0.34 Tg a$^{-1}$) with greater coal emission compared to 2017. The differences with the EDGAR v4.3.2 spatial distribution are very large. Our inventory shows spatial distributions that are broadly consistent with the NAEI, with high emissions in populated areas and production regions. Some rural areas have zero emissions in the NAEI but small (non-zero)

emissions in our inventory because of our allocation of distribution emissions by population. These areas may in fact not have access to natural gas. The NAEI has fewer offshore sources than our work because it only accounts for the offshore wells that led to the discovery of a field, rather than all wells used to exploit a field (Tsagatakis et al., 2019).

Figure 8 shows the spatial correlation coefficient of emissions between our inventory and the NAEI as a function of grid resolution. The correlation is low at 0.1° x 0.1° ($r = 0.23$) but increases rapidly as grid resolution is coarsened to 0.2° x 0.2° ($r = 0.45$), 0.5° x 0.5° ($r = 0.83$), and 1° x 1° ($r = 0.93$). At fine resolution there are slight differences in facility locations, in particular coal mines, that lead to displacement errors. There are also differences in the emissions from individual facilities reported to the NAEI that are not resolved in our inventory. These errors are rapidly smoothed out as the inventory is
averaged over a coarser grid.

## 4 Data availability

The annual gridded emission fields and gridded errors for each subsector in Fig. 1 are available on the Harvard Dataverse at https://doi.org/10.7910/DVN/HH4EUM (Scarpelli et al., 2019). Input data and code is available upon reasonable request.

## 5 Conclusions

We have constructed a global inventory of methane emissions from oil, gas, and coal with 0.1º x 0.1º resolution by spatially allocating the national emissions reported by individual countries to the United Nations Framework Convention on Climate Change (UNFCCC). The inventory differentiates oil/gas contributions from individual subsectors along the production and supply chain, and from specific processes (leakage, venting, flaring), and spatially allocates the emissions from each subsector and process using infrastructure databases. It also includes error estimates based on IPCC. Comparison with the
EDGAR v4.3.2 inventory shows large differences in terms of both national emissions and their spatial distribution. Comparison with the gridded version of the UK National Atmospheric Emissions Inventory (NAEI) shows overall good agreement but significant errors on the 0.1° x 0.1° grid that are smoothed out when our inventory is averaged on a coarser grid.

Our inventory is designed for use as prior estimate in inverse analyses of atmospheric methane observations aiming to improve knowledge of methane emissions. Corrections to emission estimates revealed by the inverse analyses can be of direct benefit to policy by identifying biases in the national inventories reported to UNFCCC. Our inventory is for 2016 but can be readily adjusted to subsequent years by updating the reported UNFCCC emissions and the Energy Information Administration's (EIA) activity data, assuming that the spatial distribution of emissions changes slowly. The validity of this
assumption will depend on the country and on the time horizon for the adjustment. In North America at least, there has been

little change over the past decade in spatial patterns of anthropogenic methane observed with the GOSAT satellite instrument (Sheng et al., 2018).

**Author Contribution**

TRS compiled data sets and created the inventory. DJJ conceived of and provided guidance for the project. JDM and JS provided guidance on North American inventories. MPS assisted in processing of spatial data. KR and LR processed and provided guidance for the infrastructure spatial data. JRW provided guidance and feedback during inventory construction. GJM was consulted for EDGAR comparison. All authors reviewed the resulting inventory and assisted with paper writing.

**Competing Interests**

The authors declare no conflict of interest.

**Acknowledgements**

This work was supported by the NASA Earth Science Division and by the NDSEG (National Defense Science and Engineering Graduate) Fellowship to TRS.

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

**Table 1. Uncertainty ranges for IPCC emission factors and corresponding error standard deviations.[a]**

| Subsector | Annex I countries | | | | Non-Annex I countries | | | |
|---|---|---|---|---|---|---|---|---|
| | Lower (%) | Upper (%) | RSD (%) | GSD | Lower (%) | Upper (%) | RSD (%) | GSD |
| Oil | | | | | | | | |
|   Exploration[b] | 100 | 100 | 50 | 2.11 | 12.5 | 800 | 100 | 1.79 |
|   Production (leakage) | 100 | 100 | 50 | 2.11 | 12.5 | 800 | 100 | 1.79 |
|   Production (venting) | 75 | 75 | 37.5 | 1.63 | 75 | 75 | 37.5 | 1.63 |
|   Production (flaring) | 75 | 75 | 37.5 | 1.63 | 75 | 75 | 37.5 | 1.63 |
|   Refining | 100 | 100 | 50 | 2.11 | 100 | 100 | 50 | 2.11 |
|   Transport (leakage) | 100 | 100 | 50 | 2.11 | 50 | 200 | 62.5 | 1.57 |
|   Transport (venting) | 50 | 50 | 25 | 1.32 | 50 | 200 | 62.5 | 1.57 |
| Gas | | | | | | | | |
|   Exploration[b] | 100 | 100 | 50 | 2.11 | 12.5 | 800 | 100 | 1.79 |
|   Production (leakage) | 100 | 100 | 50 | 2.11 | 40 | 250 | 72.5 | 1.55 |
|   Production (flaring) | 25 | 25 | 12.5 | 1.14 | 75 | 75 | 37.5 | 1.63 |
|   Processing (leakage) | 100 | 100 | 50 | 2.11 | 40 | 250 | 72.5 | 1.55 |
|   Processing (flaring) | 25 | 25 | 12.5 | 1.14 | 75 | 75 | 37.5 | 1.63 |
|   Transmission (leakage) | 100 | 100 | 50 | 1.41 | 40 | 250 | 72.5 | 1.55 |
|   Transmission (venting) | 75 | 75 | 37.5 | 1.63 | 40 | 250 | 72.5 | 1.55 |
|   Storage (leakage) | 20 | 500 | 100 | 1.65 | 20 | 500 | 100 | 1.65 |
|   Distribution (leakage) | 20 | 500 | 100 | 1.65 | 20 | 500 | 100 | 1.65 |
| Coal | 66 | 200 | 66.5 | 1.72 | 66 | 200 | 66.5 | 1.72 |

[a] The uncertainty ranges are provided by the IPCC and apply to the estimation of national emissions using emission factors specified in Tier 1 methods (IPCC, 2006). The uncertainty range for each subsector consists of an upper bound (Upper, %) and lower bound (Lower, %) on relative emissions. We interpret each uncertainty range as a 95% confidence interval, and infer the corresponding relative standard deviation (RSD, %) for the assumption of a normal error pdf and geometric standard deviation (GSD, dimensionless) for the assumption of a lognormal pdf. The IPCC provides uncertainty ranges for 'developed' and 'developing' countries and we apply them to Annex I and non-Annex I countries, respectively.

[b] Well drilling

**Table 2. Global methane emissions from oil, gas, and coal in 2016[a].**

| Sector/Subsector | Total (Tg a$^{-1}$) |
| --- | :---: |
| Oil | 41.5 |
|     Exploration | 1.4 |
|     Production (leakage) | 17.8 |
|     Production (venting) | 21.6 |
|     Production (flaring) | 0.5 |
|     Refining | 0.1 |
|     Transport (leakage) | <0.1 |
|     Transport (venting) | <0.1 |
| Gas | 24.4 |
|     Exploration | <0.1 |
|     Production (leakage) | 7.4 |
|     Production (flaring) | <0.1 |
|     Processing (leakage) | 2.3 |
|     Processing (flaring) | 0.1 |
|     Transmission (leakage) | 7.1 |
|     Transmission (venting) | 0.6 |
|     Storage (leakage) | 1.0 |
|     Distribution (leakage) | 5.7 |
| Coal | 31.3 |
| Total | 97.2 |

[a] From national totals reported to the UNFCCC with further subsector disaggregation, year adjustment, and supplemental information as given in the text.

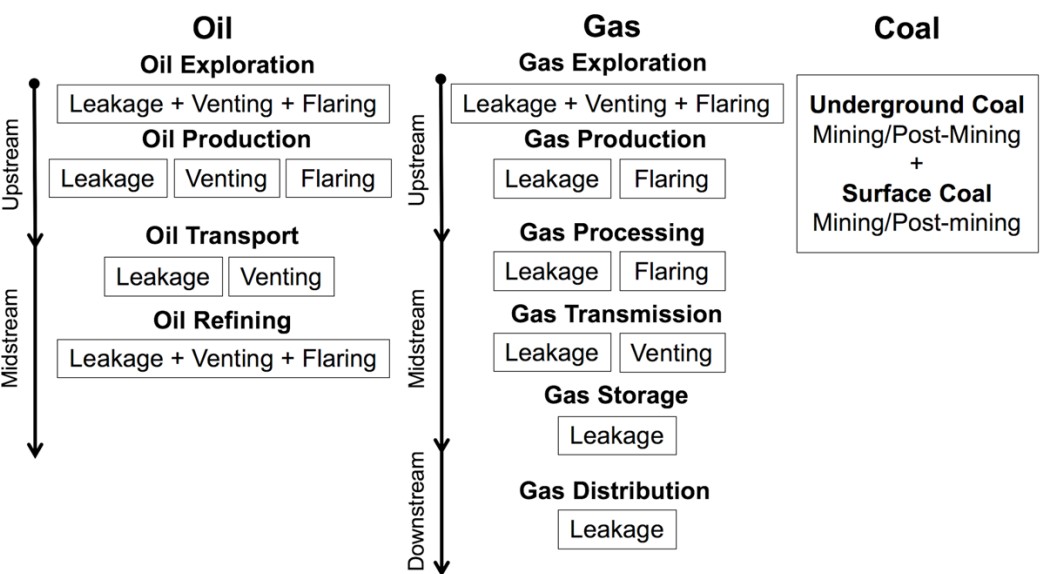

**Figure 1. Methane emissions from oil, gas, and coal as resolved in our inventory. Emissions for oil and gas are separated into subsectors representing the different lifecycle stages. Each box in the figure corresponds to a separate 0.1º x 0.1º gridded product in the inventory.**

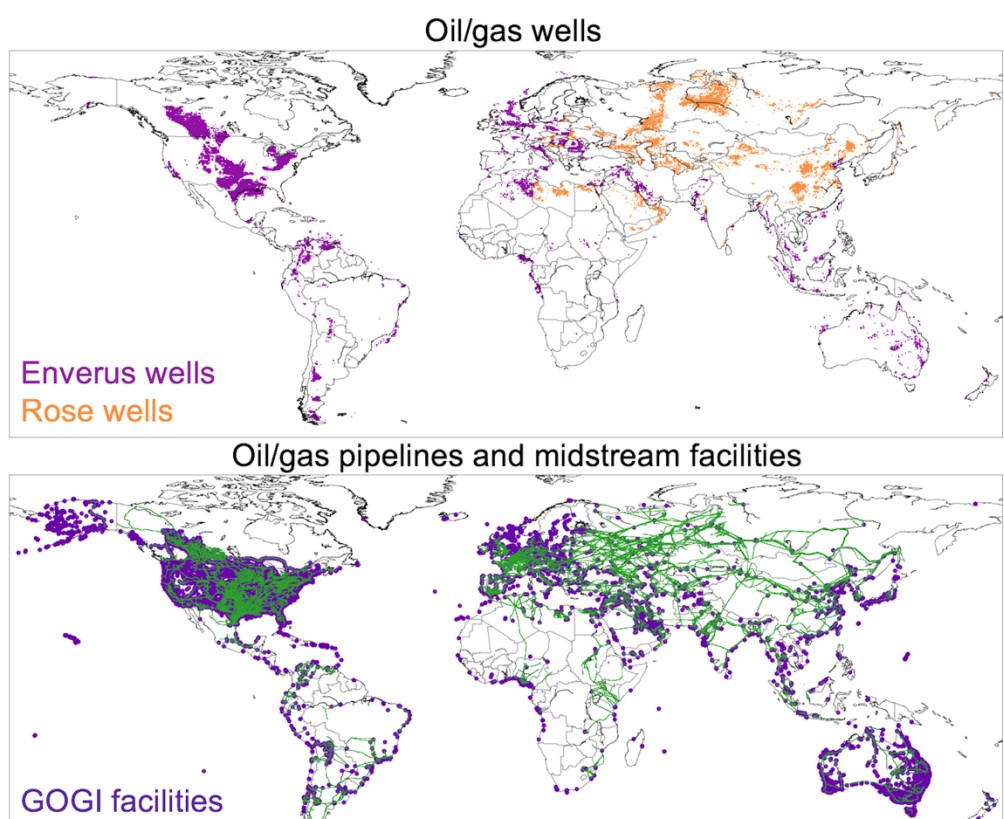

**Figure 2. Global distributions of oil/gas wells (Enverus, 2017; Rose, 2017), pipelines (EIA, 2018b; Petroleum Economist Ltd, 2010; Platts, 2008; Sabbatino et al., 2017), and midstream facilities (Sabbatino et al., 2017) used in our inventory. Well locations for individual countries are from the Enverus (2017) database where available, and from the Rose (2017) database everywhere else (see text). Wells and pipelines are gridded data at 0.1º x 0.1º grid resolution but are shown here with 0.2º x 0.2º resolution for visibility.**

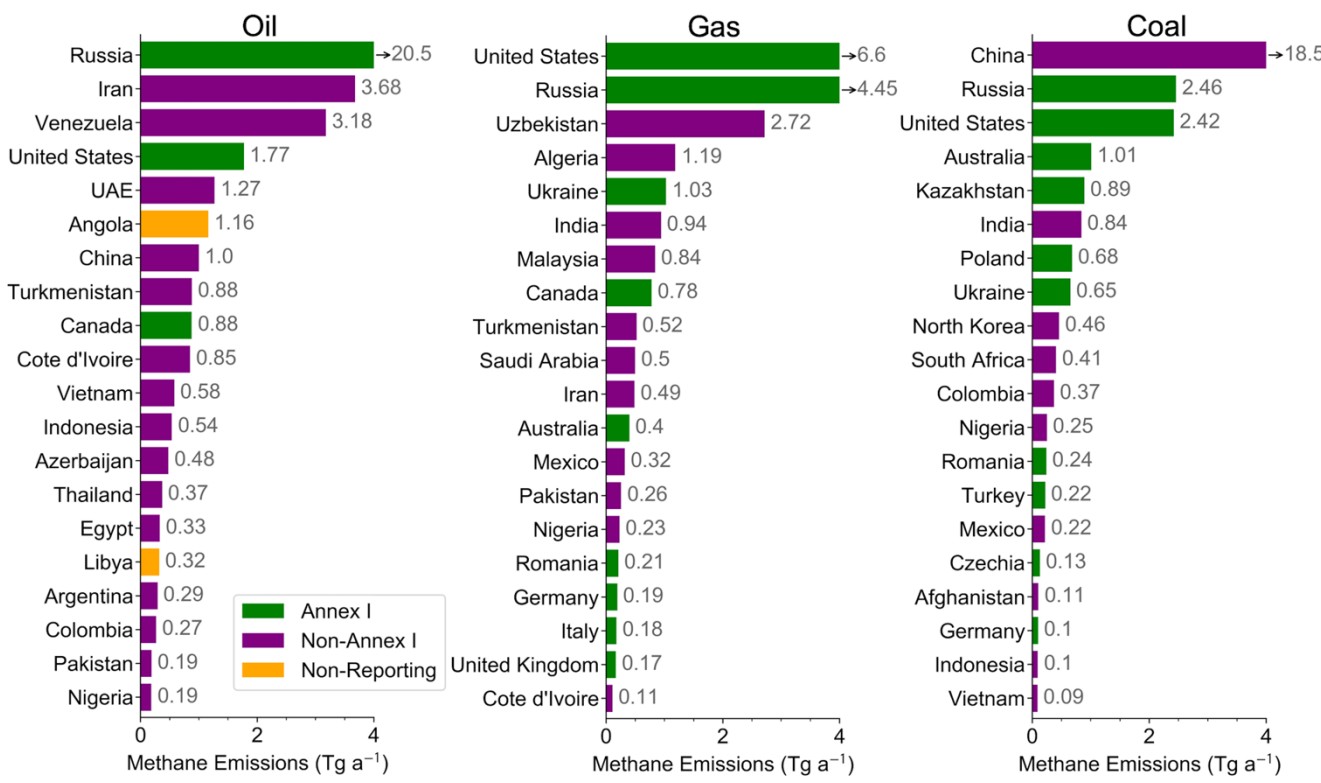

**Figure 3. Methane emissions in 2016 from the top 20 emitting countries for the oil, gas, and coal sectors. Arrows next to the top bars (highest emitting countries) indicate that emissions are not to scale.**

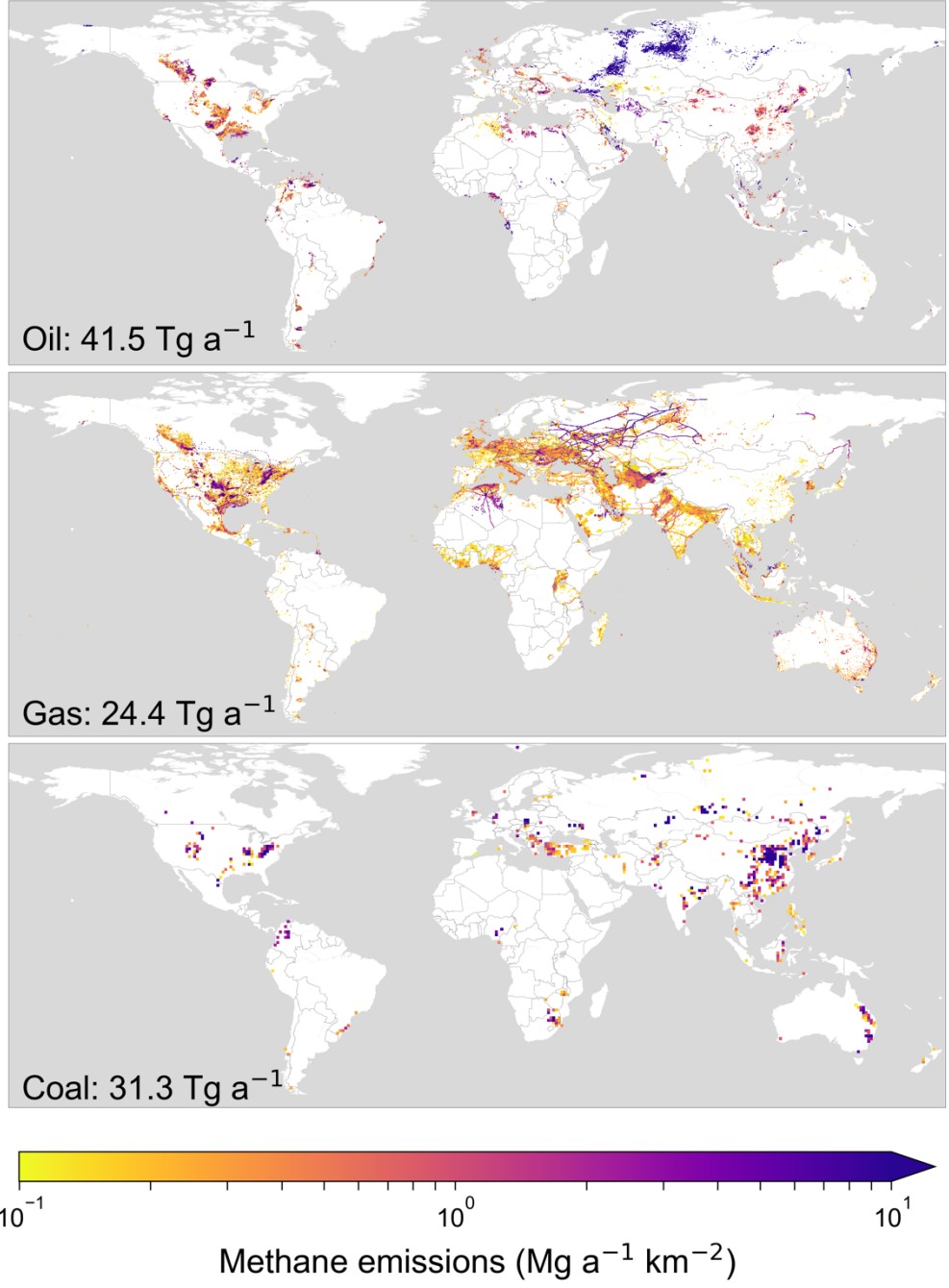

**Figure 4. Global distribution of 2016 methane emissions from oil, gas, and coal in our inventory. The inventory is at 0.1º x 0.1º grid resolution but coal is shown here at 1º x 1º resolution for visibility. Emissions below $10^{-1}$ Mg a$^{-1}$ km$^{-2}$ are not shown.**

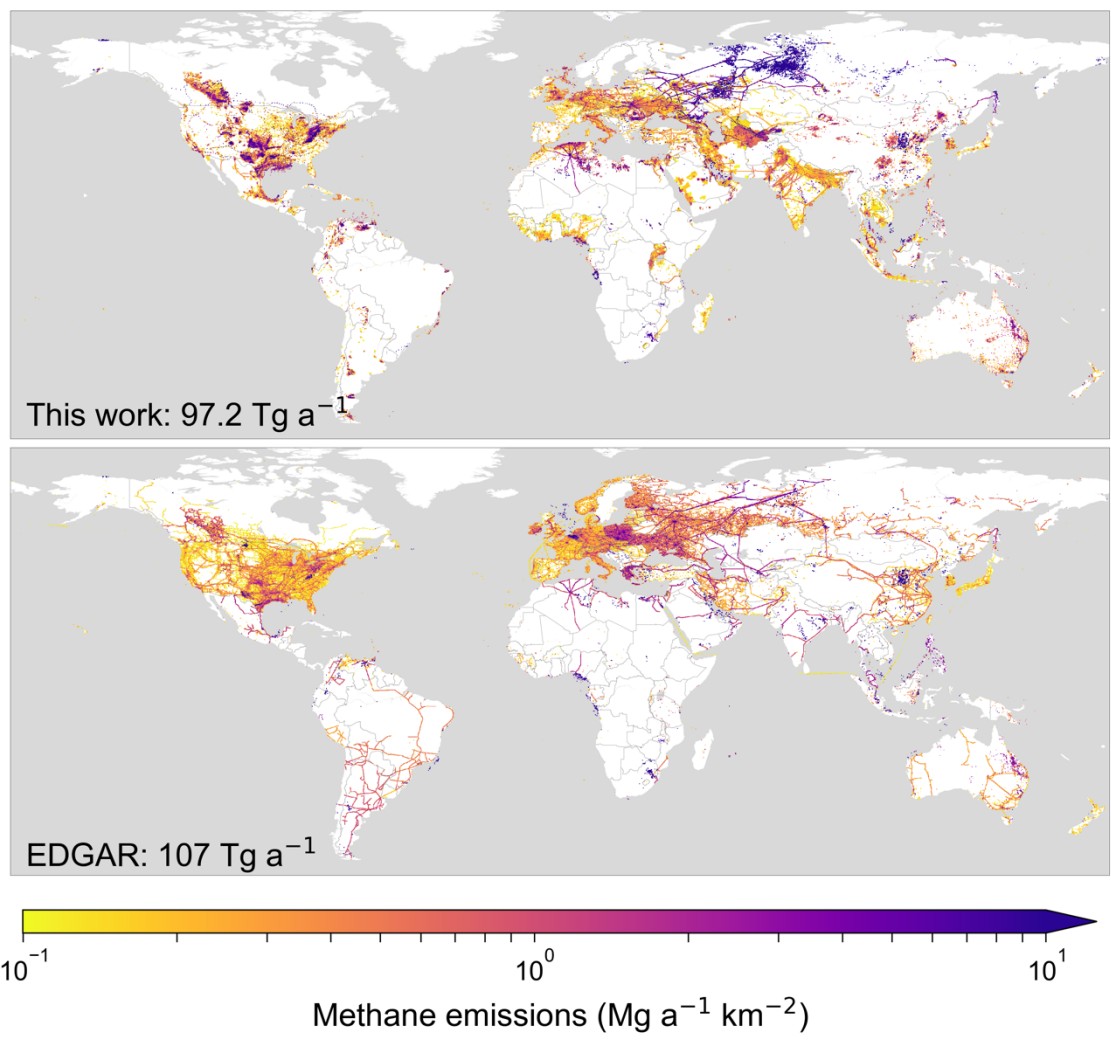

**Figure 5. Total methane emissions from fuel exploitation (sum of oil, gas, and coal) in 2016 from this work (top) and in 2012 from EDGAR v4.3.2 (bottom; European Commission, 2017). Emissions below 10⁻¹ Mg a⁻¹ km⁻² are not shown.**

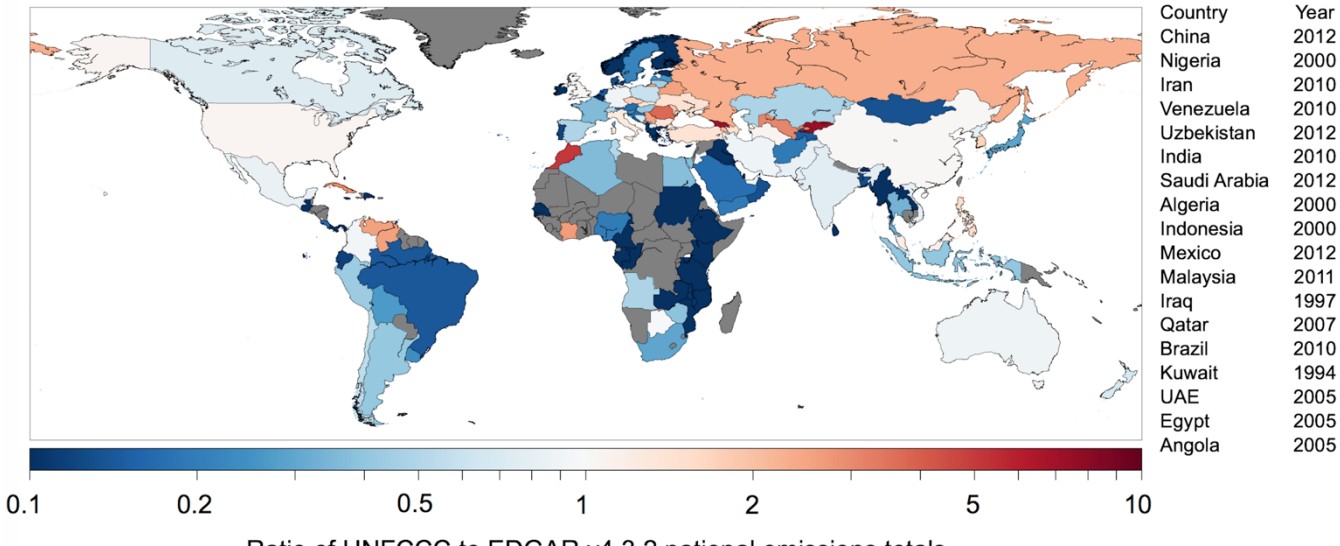

| Country | Year |
|---|---|
| China | 2012 |
| Nigeria | 2000 |
| Iran | 2010 |
| Venezuela | 2010 |
| Uzbekistan | 2012 |
| India | 2010 |
| Saudi Arabia | 2012 |
| Algeria | 2000 |
| Indonesia | 2000 |
| Mexico | 2012 |
| Malaysia | 2011 |
| Iraq | 1997 |
| Qatar | 2007 |
| Brazil | 2010 |
| Kuwait | 1994 |
| UAE | 2005 |
| Egypt | 2005 |
| Angola | 2005 |

**Figure 6. Comparison of national methane emissions from fuel exploitation (sum of oil, gas, and coal) reported by individual countries to the UNFCCC (2019) and estimated by the EDGAR v4.3.2 inventory (European Commission, 2017). The figure shows the ratio of UNFCCC to EDGAR v4.3.2 national emissions with warmer colors indicating higher UNFCCC emissions. Emissions are taken from the most recent year reported to the UNFCCC prior to or in 2012 and compared to EDGAR for the same year. The reporting year for non-Annex I countries with emissions greater than 1 Tg a⁻¹ in either inventory is shown to the right. Countries in dark grey do not report fuel exploitation emissions to the UNFCCC or have zero emissions.**

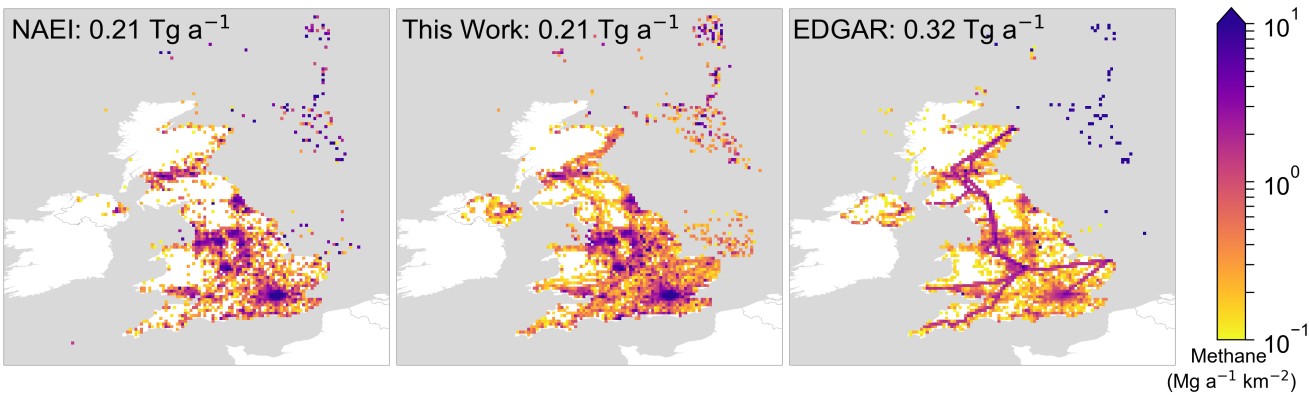

**Figure 7. Methane emissions from fuel exploitation in the United Kingdom. Our inventory (for 2016) is compared to the gridded National Atmospheric Emissions Inventory (NAEI) for 2017 (Defra and BEIS, 2019) and EDGAR v4.3.2 for 2012 (European Commission, 2017). Emissions below 10⁻¹ Mg a⁻¹ km⁻² are not shown. National total emissions are given inset. We have masked EDGAR offshore emissions using the other two inventories.**

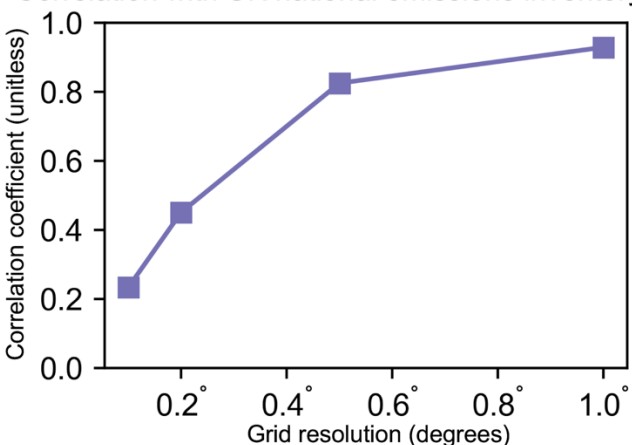

**Figure 8. Spatial correlation between gridded United Kingdom emissions in our inventory and in the National Atmospheric Emissions Inventory (NAEI; Defra and BEIS, 2019). The figure shows the Pearson correlation coefficient ($r$) at the native 0.1° x 0.1° grid resolution of our inventory and after averaging over coarser grid resolutions up to 1° x 1°.**