# Peer review of "A global gridded $(0.1^{\circ} \times 0.1^{\circ})$ inventory of methane emissions from oil, gas, and coal exploitation based on national reports to the United Nations Framework Convention on Climate Change"

_Earth System Science Data, 2019_

## Referee Comment (RC1) · Anonymous Referee #1 · 4 Oct 2019

Review of ESSD-2019-127:
**A global gridded (0.1° x 0.1°) inventory of methane emissions from oil, gas, and coal exploitation based on national reports to the United Nations Framework Convention on Climate Change**

This paper presents a new gridded inventory of methane emissions from the oil, gas and coal sectors for 2016. This paper does not present large scientific developments but rather acts as a companion to the emissions dataset, which is publicly available. The paper and dataset represent a significant advance for bottom up inventories and prior model emission estimates, and as such this is ideal for publication in Earth System Science Data. The manuscript is well written and clear, and figures are high quality. I recommend publication following minor revisions as described below.

**Comments:**
- P2 L17-20: You list these other gridded CH4 inventories here but do not seem to compare to them later. I would like to see a comparison to at least some of them added, since at the moment you only compare to EDGAR, which is not sectorally resolved.
- P3 L19: Why did you choose the year 2016? Comment on 2016 emissions in the context of interannual emission trends in emissions eg. was it a particularly low/high/normal year? Can you add a figure showing interannual emissions from these sectors and perhaps subsectors also for a relevant time range eg. 2000-2019?
- P4 L3-5 seems to relate to P3 L30 with other information in between – this paragraph could be restructured to present the information more clearly.
- P4 L27-31: You say "notably" to one or two countries for each case here. Are these the only countries for each case? Or the only countries with emissions above a certain level? If the latter, what cut off level did you use to define "notably"?
- P5 L10-11: How old were the estimates in the US inventory? Is the balance between sectors likely to have changed eg. as the energy landscape has changed? Did you need to account for this at all?
- P5 L18-19: DrillingInfo.com is now enverus.com. Change in your reference list so that the link continues to be valid but add info also relevant to when you accessed it.
- P6 L3-4: Again you use "notably" – please state more clearly how you defined this, eg. "the top five emitters in this category were…" or similar.
- Section 2.3: This is not my area of expertise but forms an important part of this paper. If the second reviewer is also not expert in this area I would consider getting the opinion of a third reviewer on error treatment within this dataset.
- P8 L25-26: similar…to
- P8 L28: Why would the Global Carbon Project bottom up estimates be so much higher than your estimate? What differences are there in the calculation approach that would cause this?
- P10 L19: How valid is this assumption, ie. what do you mean by "slowly"? Can you give a time and error range? Perhaps using activity data for 2013 emissions is fine but 1990 or 2030 emission spatial distribution may not be similar.

- Figure 1: It would be nice if you could combine in this a schematic image of each sector, to give a visual representation of the different stages contributing emissions.
- Figure 3: Add the cumulative % of emissions for each sector to this figure.
- Figure 4 and 5: The grey background makes the figure harder to interpret; please change to a white background like Figure 2.

---

## Referee Comment (RC2) · Anonymous Referee #2 · 9 Oct 2019

Review of "A global gridded (0.1ïĆřïĆť0.1ïĆř) inventory of methane emissions from oil, gas, and coal exploitation based on national reports to the United Nations Framework Convention on Climate Change" by Scarpelli et al. (ESSD-2019-127).

The paper describes the global inventory of methane emissions from oil, gas, and coal for 2016. The gridded emission data set is publicly available. Their inventory resolves the subsectors of oil and gas exploitation, from upstream to downstream, and

the different emission processes. The study also gives a gridded error estimate based on emission factor uncertainties from the IPCC. The inventory is compared with the EDGAR v4.3.2 global inventory. The work is an advancement in methane emissions inventory and the paper can be considered for publication in Earth System Science Data after addressing the following minor comments.

Comments: Page 2 lines 14-15: What are some of the other regional and global multi-species emission inventories? Suggest naming a few. Page 2 line 30: Why was the year 2016 chosen for the study? Page 4 line 9: The emission inventories over the US are for which year? Page 5 line 19: Drillinginfo is now enverus. Please make the changes to the manuscript. Page 6 lines 28-33: Is the refining rate threshold based on the largest refinery? Does it mean the same for processing plants, storage facilities, and compressor stations? Page 8 line 32 Page 9 line 1: What do the authors mean to say in the line 'Oil and gas emissions . . . of the two fuels'? Figures 4 and 5: Replace the gray background with white for clarity.

---

## Short Comment (SC1) · 17 Oct 2019

I have only a few issues for the main text:

- Assuming a normal distribution, the 95% confidence interval has a range of 4-$\sigma$ ($\pm$ 2-$\sigma$).

- Allocating the errors from national scale to gridcell scale should be done taking

care of uncertainty propagation. This is always very unclear in other studies so I believe consistency between the national and gridcell scale are very important.

- It would be useful to include uncertainty estimates in table 2 (both for normal and log-normal distributions).

- The information for coal seems to be a stump. It would very useful if you would include separate emissions for underground and surface mining, post mining operations and type of coal (at least lignite vs bituminous or anthracite) as these data can be of use for isotope studies, e.g. Zazzerri et al. (2016)

- Figure 2: it would be useful to also see locations of refineries, storage stations, gas processing stations, etc.

- Figure 3: Comparison to other inventories would be useful. As would be maybe a latitudinal profile of emissions between the different inventories.

One of the main novel issues of these study is the more transparent use of an array of databases is used to spatially allocate national emissions to infrastructure including wells, pipelines, oil refineries, gas processing plants, gas compressor stations, gas storage facilities, and coal mines. However, when looking at the netcdf data explicitly, I found a number of inconsistencies in the allocation of emissions:

- Emissions from gas processing (both flaring and fugitive emissions) where allocated to pipelines in Eurasia, Northern Africa, and South America. For flaring, there is no data for the US.

- Emissions from gas storage in North America and parts of Asia are also distributed to pipelines.

- Emissions from venting during gas transmission is not given for several countries including Russia and the US, which are the most important in this sector.

- Emissions from oil refining and transport in North America were allocated to oil fields and not to the actual refineries location or pipelines. The emissions from oil tankers found in EDGAR are not found in this inventory.

- Coal emissions in Japan, Netherlands and Ireland are distributed according to population, whereas the location of such mines are quite specific in these countries, e.g. https://en.wikipedia.org/wiki/List_of_mines_in_Ireland, https://www.tudelft.nl/en/ceg/about-faculty/departments/geoscience-engineering/sections/resource-engineering/links/coal-mining-in-the-netherlands/ and https://en.wikipedia.org/wiki/List_of_coal_mines_in_Japan.

- No distinction seems to be made between oil and gas wells and pipelines, such that the distribution production emissions from both types of fossil fuels is very similar.

From my own research I can point to several other datasets that help with the allocation of emissions:

- Flaring
  - The radiant output of gas flares contained in the Worldwide natural gas flaring dataset [http://skytruth.org/viirs/] (free data but only for one year).
  - The gold standard would be the data from Elvidge et al. (2007, 2009, 2012) but it seems not to be publicly available.

- Oil storage: https://tankterminals.com

- Oil pipelines: http://worldmap.harvard.edu/data/geonode:global_oil_pipelines_7z9

- Refineries: Oil & Gas Journal periodic Worldwide Refining Survey

- Oil vs Gas fields:
    - Petroleum Dataset v. 1.2 (Lujala, 2007)
    - Giant Oil and Gas Fields of the World database (Horn, 2004)
    - US shale plays (EIA, 2011)

References:

Elvidge, C., Baugh, K. E., Tuttle, B. T., Howard, A. T., Pack, D. W., Milesi, C., Erwin, E. H. (2007). A Twelve Year Record of National and Global Gas Flaring Volumes Estimated Using Satellite Data (pp. 1–107). World Bank.

Elvidge, C. D., Ziskin, D., Baugh, K. E., Tuttle, B. T., Ghosh, T., Pack, D. W., et al. (2009). A Fifteen Year Record of Global Natural Gas Flaring Derived from Satellite Data. Energies, 2(3), 595–622. http://doi.org/10.3390/en20300595

Elvidge, C., Zhizhin, M., Baugh, K. E., Hsu, F.-C., Ghosh, T. (2016). Methods for Global Survey of Natural Gas Flaring from Visible Infrared Imaging Radiometer Suite Data. Energies, 9(1), 14. http://doi.org/10.3390/en9010014

Horn, M. K. (2004). Selected Features of Giant Fields, Using Maps and Histograms. American Association of Petroleum Geologist Search and Discovery, 1–14. Retrieved from http://www.searchanddiscovery.com/documents/2004/horn/index.htm

Lujala, P., d, J. R., Thieme, N. (2007). Fighting over Oil: Introducing a New Dataset. Conflict Management and Peace Science, 24(3), 239–256. http://doi.org/10.1080/07388940701468526

US Energy Information Adminstration, I. S. A. (2011, May 9). North American shale plays. International Energy Statistics. Retrieved from https://www.eia.gov/maps/maps.htmgeodata

Zazzeri, G., Lowry, D., Fisher, R. E., France, J., Lanoisellé, M., Kelly, B. F. J., et al.
(2016). Carbon isotopic signature of coal-derived methane emissions to atmosphere: from coalification to alteration. Atmospheric Chemistry and Physics, 16, 13669–13680. http://doi.org/10.5194/acp-16-13669-2016

―――――――――――――――――――

---

## Short Comment (SC2) · 12 Nov 2019

Dear Editor, Dear authors,

We are contacting you regarding the afore mentioned article on discussion phase for publication.

As part of the findings, results are compared with publicly available data from EDGAR

4.3.2; highlighting differences between EDGAR, their inventory and national submissions to the United Nations Framework Convention on Climate Change.

First, we would like to inform the authors that an updated version of the 0.1° x 0.1° gridded EDGAR global greenhouse emission inventory has recently been released – including CO2 emissions up to 2018 and non-CO2 greenhouse gases up to 2015. Data for this new release (EDGAR v5.0_FT2018) can be accessed through our website: https://edgar.jrc.ec.europa.eu/overview.php?v=50_GHG.

Users can also download the "2019 Global Fossil CO2 and Greenhouse Gas Emission report" (Crippa et al., 2019) , in which the most relevant methodological aspects for emission quantification from large emitting sectors and trends for large emitting countries are described. We also expect to publish an article in the coming months; analysing the role of socio-economic transitions and GHG mitigation policies on global emission trends (Oreggioni et al.,in preparation).

Given that CH4 fugitive emissions from fuel supply chain are the subject of undergoing discussion in the scientific community and in the policy making decision process, the EDGAR v5.0_FT2018 disaggregated fuel-based fugitive emissions are openly accessible from our site.

We have the following comments concerning the differences between EDGAR quantified emissions, this work and nationally reported data to the UNFCC:

- As the authors point out, there is significant uncertainty in the estimation of CH4 fugitive emissions from the different stages of the fuel supply chain. CH4 emissions from the exploration phase, the production facilities, flaring and fuel transport depend on: well characteristics, chemical and physical properties of the extracted fuel, operating conditions of the processes in the fuel production plant, pipeline design and ambient conditions respectively. These site-specific characteristics affect the representativeness of the emission factors; specially if Tier 1/ default IPCC factors are considered.

- EDGAR is an independent tool that uses a consistent and transparent methodology for the estimation of greenhouse gas and air pollutant emissions for all the IPCC categories for all countries. EDGAR aims to use the most representative and up to date activity data, technology matrix and emission factors. However, given EDGAR global coverage, assumptions based on regional similarity or Tier 1 default emission factors may be needed for emission quantification; particularly for non-combustion sectors in Non-Annex I countries.

- Emissions from all IPCC categories are included in the EDGAR database. We frequently present aggregated values on our website, but the level of detail in the calculation is that recommended by the 2006 IPCC Guidelines. Attached, you can find a summary of the sources for the activity data, emission factors and methods for spatial disaggregation of emissions associated with the different IPCC sub-categories for CH4 fugitive emissions. Further details can be found in Janssens-Maenhout et al. (2019 ) and Crippa et al. (2019). This may help the authors understand the differences between the approaches used in our database for the emission quantification, the methodology presented in the manuscript and national submissions to UNFCCC. Finally, we would like to remind the authors and other EDGAR users, that we can also be contacted either directly or via the functional e-mail JRC-EDGAR@ec.europa.eu, in case of questions or requests of information that is not available on our website or in our published papers and reports. We are committed to respond to queries as quickly as possible; providing a more detailed explanation of our methodology and more disaggregated results if needed, subject to any intellectual property agreements with our data suppliers.

Kind regards,

M. Crippa, G. Oreggioni, M. Muntean, F. Monforti-Ferraio, E. Schaaf, D. Guizzardi, M.Duerr

Cited references

Crippa, M., Oreggioni, G., Guizzardi, D., Muntean, M.;Schaaf, E., LoVullo, E..,Solazzo, E..,Monforti-Ferrario, F.; Olivier, J.G.J.; Vignati, E.: Fossil CO2 and GHG emissions of all world countries – 2019 Report, EUR 29849 EN, Publications Office of the European Union, Luxembourg, 2019, ISBN 978-92-76-11100-9, doi:10.2760/687800, JRC117610.

Oreggioni, G., Crippa, M., Monforti-Ferrario, F., Muntean, M., LoVullo, E., Schaaf, E.,Guizzardi, D.,Solazzo, E., Oliver, J G J, Vignati E. Using EDGAR v5.0 to understand the role of socio-economic transition and mitigation policies on global GHG emissions. In preparation.

Janssens-Maenhout, G., Crippa, M., Guizzardi, D., Muntean, M., Schaaf, E., Dentener, F., Bergamaschi, P., Pagliari, V., Olivier, J. G. J., Peters, J. A. H. W., van Aardenne, J. A., Monni, S., Doering, U., Petrescu, A. M. R., Solazzo, E., and Oreggioni, G. D.: EDGAR v4.3.2 Global Atlas of the

Please also note the supplement to this comment:
https://www.earth-syst-sci-data-discuss.net/essd-2019-127/essd-2019-127-SC2-supplement.pdf

**Supplement:**

*Table: Described methodology for emission quantification and spatial allocation for CH$_4$ fugitive emission[1]s.*

| IPCC Code | EDGAR sector | Activity Data | Emission factor | Spatial disaggregation proxy |
|---|---|---|---|---|
| **1B1 (Solid fuel)** | | | | |
| **1B 1 (Mining)** | **Brown coal** PRO.BRC.LGN PRO.BRC.NSF

 **Hard coal** PRO.HDC.ANT PRO.HDC.BTC PRO.HDC.CKC PRO.HDC.SBC | Coal production statistics using data from World Coal Association data. | IPCC Tier 1EF (IPCC, 2006) | Combining USGS coal mines (https://www.usgs.gov/), EPRTRv4.2 for European mines (http://prtr.ec.europa.eu) and Global Energy Observatory ( http://globalenergyobservatory.org) and China coal mine data from Liu et al. (2015). |
| **1B1 (Post-mining and CH4 recovery)** | **Brown coal** PRO.BRC.PST

 **Hard coal** PRO.HDC.PST PRO.HDC.REC | | | |
| **1B2 a (Oil and natural gas supply value chain)** | | | | |
| **Exploration** | **Natural gas** PRO.GAS.NGS **Oil** PRO.GAS.NGS,PRO.OIL.BDS PRO.OIL.BGL,PRO.OIL.CNF | Gas and oil production data, based on IEA fuel balances (2017) | IPCC Tier 1EF (IPCC, 2006) | NOAA-NDGC (2015) |
* * *
[1] Further details can be found in Janssens-Maenhout (2019)

| | | | | |
|---|---|---|---|---|
| | PRO.OIL.CRU, PRO.OIL.NCR, PRO.OIL.NGL, PRO.OIL.OPR | | | |
| **Transport and distribution** | *Natural gas* PRO.GAS.DIS, PRO.GAS.PIP *Oil* PRO.OIL.PIP, PRO.OIL.TK4 | Rate calculated based on length of pipe and material –using data from: Eurogas (2010), Marcogaz (2013), submissions to UNFCC and CIA (2016). | IPCC Guidelines (2006) supplemented with data from country submissions to UNFCCC and Lelieveld et al. (2005). | **Gas transmission and oil transmission**: Janssens-Maenhout et al.(2014)) **Oil terminals**: World Port (2015), **Oil tankers:** Trombetti et al. (2017); Wang et al. (2007). **Gas pipeline distribution**: proxies developed using population CIESIN (2011) and road types Geofabrik (2015). |
| **Flaring** | PRO.GAS.VAF | Statistics for the total amount of gas flared and vented using data from EIA and national submissions to UNFCCC. The share that is flared was calculated using the NOAA satellite observation of flaring lights NOA-NGDC. | **Flaring:** EF Based on national submissions to UNFCCC. **Venting:** IPCC Tier 1 EF (IPCC, 2006) | NOAA-NDGC (2015) Elvidge et al. (2016) |

---

## Author Comment (AC1) · 20 Dec 2019

**Response to Anonymous Referee 1 (RC1)**

We would like to thank the reviewer for the positive evaluation and for the useful feedback. Responses to comments are posted below the relevant comment. Referee comments are italicized.

*P2 L17-20: You list these other gridded CH4 inventories here but do not seem to compare to them later. I would like to see a comparison to at least some of them added, since at the moment you only compare to EDGAR, which is not sectorally resolved.*

Thank you for this suggestion. We have added a comparison to the UK gridded inventory to the revised manuscript.

*P3 L19: Why did you choose the year 2016? Comment on 2016 emissions in the context of interannual emission trends in emissions eg. was it a particularly low/high/normal year? Can you add a figure showing interannual emissions from these sectors and perhaps subsectors also for a relevant time range eg. 2000-2019?*

This year was chosen because at the time of inventory construction it was the most recent year available for Annex I national emissions from the UNFCCC. We have clarified this in the text. The UNFCCC (https://di.unfccc.int/detailed_data_by_party) allows you to access data regarding interannual emissions trends.

*P5 L10-11: How old were the estimates in the US inventory? Is the balance between sectors likely to have changed eg. as the energy landscape has changed? Did you need to account for this at all?*

Yes, it is likely that in the US there have been changes to the energy landscape from the US inventory year in 2012 to 2016. The scaling we mention in the text is done by subsector (we have added this to the text) so subsector emissions and the balance between sectors will match the national emissions reported to the UNFCCC for 2016. As the US EPA inventory is updated we will be able to substitute the new EPA gridded inventory for the Maasakkers et al. 2016 version to account for any spatial changes in emission sources.

*P8 L28: Why would the Global Carbon Project bottom up estimates be so much higher than your estimate? What differences are there in the calculation approach that would cause this?*

We have removed this sentence from the text because the bottom-up estimates are based primarily on outdated inventories including EDGAR v4.2 and the US EPA which uses outdated national emissions reported to the UNFCCC.

*P10 L19: How valid is this assumption, ie. what do you mean by "slowly"? Can you give a time and error range? Perhaps using activity data for 2013 emissions is fine but 1990 or 2030 emission spatial distribution may not be similar.*

The rate of spatial change in emissions and the degree of the spatial variability will be very country specific, so it is hard to make any general statement. We have added a statement to that effect in the introduction of the text.

*P4 L3-5 seems to relate to P3 L30 with other information in between – this paragraph could be restructured to present the information more clearly.*

Done.

*P4 L27-31: You say "notably" to one or two countries for each case here. Are these the only countries for each case? Or the only countries with emissions above a certain level? If the latter, what cut off level did you use to define "notably"?*

Clarified.

*P5 L18-19: DrillingInfo.com is now enverus.com. Change in your reference list so that the link continues to be valid but add info also relevant to when you accessed it.*

Done.

*P6 L3-4: Again you use "notably" – please state more clearly how you defined this, eg. "the top five emitters in this category were..." or similar.*

Clarified.

*P8 L25-26: similar...to*

Corrected.

**Response to Anonymous Referee 2 (RC2)**

We would like to thank the reviewer for the positive evaluation and for the useful feedback. Responses to comments are posted below the relevant comment. Referee comments are italicized.

*Page 2 lines 14-15: What are some of the other regional and global multi- species emission inventories? Suggest naming a few.*

We now name the inventories.

*Page 2 line 30: Why was the year 2016 chosen for the study?*

This year was chosen because at the time of inventory construction it was the most recent year available for Annex I national emissions from the UNFCCC. We have clarified this in the text.

*Page 6 lines 28-33: Is the refining rate threshold based on the largest refinery? Does it mean the same for processing plants, storage facilities, and compressor stations?*

Yes, an effort was made to find the largest processing plant and refinery. This is now stated. For storage, as we state in the text we simply use the US storage capacity and for compressor stations we use the upper limit of expected distances between stations. These are conservative thresholds and we have clarified this in the text. Our goal is to avoid false "hotspot" emissions at facilities rather than accurately estimate if facilities are missing. We now say this in the text.

*Page 5 line 19: Drillinginfo is now enverus. Please make the changes to the manuscript.*

Done.

*Page 4 line 9: The emission inventories over the US are for which year?*

Clarified.

*Page 8 line 32 Page 9 line 1: What do the authors mean to say in the line 'Oil and gas emissions . . . of the two fuels'?*

Clarified.

**Response to Short Comment 1 (**Tonatiuh Guillermo Nuñez Ramirez)

Thank you to Tonatiuh Guillermo Nuñez Ramirez for providing short comments. We found the comments helpful and have added a map of facilities to Figure 3 and a comparison with the UK gridded inventory to Figure 7. We also thank you for the suggestion of other possible data sources and will consider them for incorporation in any future updated versions of the inventory.

**Response to Short Comment 2 (**Gabriel Oreggioni)

Thank you to Gabriel Oreggioni for pointing out the new version of EDGAR and for providing additional details on this dataset. It will be useful to compare the fuel specific emissions available with EDGAR v5.0 with our work in the future. We now cite EDGAR v5.0 as the most recent version available.

---

## Author Response (AR2)

**Response to Anonymous Referee 1 (RC1)**

We would like to thank the reviewer for the positive evaluation and for the useful feedback. Responses to comments are posted below the relevant comment. Referee comments are italicized.

*P2 L17-20: You list these other gridded CH4 inventories here but do not seem to compare to them later. I would like to see a comparison to at least some of them added, since at the moment you only compare to EDGAR, which is not sectorally resolved.*

Thank you for this suggestion. We have added a comparison to the UK gridded inventory to the revised manuscript.

*P3 L19: Why did you choose the year 2016? Comment on 2016 emissions in the context of interannual emission trends in emissions eg. was it a particularly low/high/normal year? Can you add a figure showing interannual emissions from these sectors and perhaps subsectors also for a relevant time range eg. 2000-2019?*

This year was chosen because at the time of inventory construction it was the most recent year available for Annex I national emissions from the UNFCCC. We have clarified this in the text. The UNFCCC (https://di.unfccc.int/detailed_data_by_party) allows you to access data regarding interannual emissions trends.

*P5 L10-11: How old were the estimates in the US inventory? Is the balance between sectors likely to have changed eg. as the energy landscape has changed? Did you need to account for this at all?*

Yes, it is likely that in the US there have been changes to the energy landscape from the US inventory year in 2012 to 2016. The scaling we mention in the text is done by subsector (we have added this to the text) so subsector emissions and the balance between sectors will match the national emissions reported to the UNFCCC for 2016. As the US EPA inventory is updated we will be able to substitute the new EPA gridded inventory for the Maasakkers et al. 2016 version to account for any spatial changes in emission sources.

*P8 L28: Why would the Global Carbon Project bottom up estimates be so much higher than your estimate? What differences are there in the calculation approach that would cause this?*

We have removed this sentence from the text because the bottom-up estimates are based primarily on outdated inventories including EDGAR v4.2 and the US EPA which uses outdated national emissions reported to the UNFCCC.

*P10 L19: How valid is this assumption, ie. what do you mean by "slowly"? Can you give a time and error range? Perhaps using activity data for 2013 emissions is fine but 1990 or 2030 emission spatial distribution may not be similar.*

The rate of spatial change in emissions and the degree of the spatial variability will be very country specific, so it is hard to make any general statement. We have added a statement to that effect in the introduction of the text.

*P4 L3-5 seems to relate to P3 L30 with other information in between – this paragraph could be restructured to present the information more clearly.*

Done.

*P4 L27-31: You say "notably" to one or two countries for each case here. Are these the only countries for each case? Or the only countries with emissions above a certain level? If the latter, what cut off level did you use to define "notably"?*

Clarified.

*P5 L18-19: DrillingInfo.com is now enverus.com. Change in your reference list so that the link continues to be valid but add info also relevant to when you accessed it.*

Done.

*P6 L3-4: Again you use "notably" – please state more clearly how you defined this, eg. "the top five emitters in this category were..." or similar.*

Clarified.

*P8 L25-26: similar...to*

Corrected.

**Response to Anonymous Referee 2 (RC2)**

We would like to thank the reviewer for the positive evaluation and for the useful feedback. Responses to comments are posted below the relevant comment. Referee comments are italicized.

5  *Page 2 lines 14-15: What are some of the other regional and global multi- species emission inventories? Suggest naming a few.*

We now name the inventories.

10  *Page 2 line 30: Why was the year 2016 chosen for the study?*

This year was chosen because at the time of inventory construction it was the most recent year available for Annex I national emissions from the UNFCCC. We have clarified this in the text.

15  *Page 6 lines 28-33: Is the refining rate threshold based on the largest refinery? Does it mean the same for processing plants, storage facilities, and compressor stations?*

Yes, an effort was made to find the largest processing plant and refinery. This is now stated. For storage, as we state in the text we simply use the US storage capacity and for compressor stations we use the upper limit of expected distances between

20  stations. These are conservative thresholds and we have clarified this in the text. Our goal is to avoid false "hotspot" emissions at facilities rather than accurately estimate if facilities are missing. We now say this in the text.

*Page 5 line 19: Drillinginfo is now enverus. Please make the changes to the manuscript.*

25  Done.

*Page 4 line 9: The emission inventories over the US are for which year?*

Clarified.

*Page 8 line 32 Page 9 line 1: What do the authors mean to say in the line 'Oil and gas emissions . . . of the two fuels'?*

Clarified.

**Response to Short Comment 1 (**Tonatiuh Guillermo Nuñez Ramirez)

We would like to thank Tonatiuh Guillermo Nuñez Ramirez for providing short comments. Each comment is italicized below with point by point responses posted after the relevant comment.

*I have only a few issues for the main text:*

- *Assuming a normal distribution, the 95% confidence interval has a range of 4-σ (± 2-σ)*

Clarified in Section 2.3.

- *Allocating the errors from national scale to gridcell scale should be done taking care of uncertainty propagation. This is always very unclear in other studies so I believe consistency between the national and gridcell scale are very important.*

15  Clarified in Section 2.3 and Table 1.

- *It would be useful to include uncertainty estimates in table 2 (both for normal and log-normal distributions).*

We do not include uncertainty estimates with the global emission sums because the covariance structure needed to aggregate
20  national uncertainties is unknown.

- *The information for coal seems to be a stump. It would very useful if you would include separate emissions for underground and surface mining, post mining operations and type of coal (at least lignite vs bituminous or anthracite) as these data can be of use for isotope studies, e.g. Zazzerri et al. (2016)*

We do not have separate spatial datasets for different types of coal, but if these become available at the global scale we will consider incorporating them into future versions of the inventory.

- *Figure 2: it would be useful to also see locations of refineries, storage stations, gas processing stations, etc.*

Done.

- *Figure 3: Comparison to other inventories would be useful. As would be maybe a latitudinal profile of emissions between the different inventories.*

35

We found this suggestion very helpful and have added a comparison with the UK gridded inventory (see Section 3.3, Figure 7, and Figure 8).

*One of the main novel issues of these study is the more transparent use of an array of databases is used to spatially allocate national emissions to infrastructure including wells, pipelines, oil refineries, gas processing plants, gas compressor stations, gas storage facilities, and coal mines. However, when looking at the netcdf data explicitly, I found a number of inconsistencies in the allocation of emissions:*

- *Emissions from gas processing (both flaring and fugitive emissions) where allocated to pipelines in Eurasia, Northern Africa, and South America. For flaring, there is no data for the US.*

Gas processing emissions are allocated to pipelines in some regions to account for any processing plants that are missing from our spatial datasets. We have clarified in the text that some Annex I countries report "Included Elsewhere" for venting and flaring emissions and this is the case for the US.

- *Emissions from gas storage in North America and parts of Asia are also distributed to pipelines.*

Gas storage emissions are allocated to pipelines in some regions to account for any missing gas storage facilities from our spatial datasets.

- *Emissions from venting during gas transmission is not given for several countries including Russia and the US, which are the most important in this sector.*

The US and Russia report gas venting emissions as "Included Elsewhere".

- *Emissions from oil refining and transport in North America were allocated to oil fields and not to the actual refineries location or pipelines. The emissions from oil tankers found in EDGAR are not found in this inventory.*

The Sheng et al. (2017) and Maasakkers et al. (2016) inventories only provide a total oil emissions gridded product which we have clarified in the text. For our oil transport we assume emissions primarily occur during product transfers close to pipeline infrastructure.

- *Coal emissions in Japan, Netherlands and Ireland are distributed according to population, whereas the location of such mines are quite specific in these countries, e.g. https://en.wikipedia.org/wiki/List_of_mines_in_Ireland, https://www.tudelft.nl/en/ceg/about-faculty/departments/geoscience-engineering/ sections/resource-engineering/links/coal-mining-in-the-netherlands/ and https://en.wikipedia.org/wiki/List_of_coal_mines_in_Japan.*

We have used EDGAR v4.3.2 for spatial allocation of coal emissions due to its global coverage. For future inventory updates we will use the most recent version of EDGAR which has improved coal mine data or additional global datasets if available.

- *No distinction seems to be made between oil and gas wells and pipelines, such that the distribution production emissions from both types of fossil fuels is very similar.*

When wells have both oil and gas listed as a resource or when we do not have information on the primary resource being produced we must allocate both oil and gas emissions to the same wells leading to similar oil/gas emissions footprints. For some pipelines in our dataset the resource was unknown so both oil and gas emissions were designated to the pipeline.

*From my own research I can point to several other datasets that help with the allocation of emissions…*

We thank you for the suggestion of other possible data sources and will consider them for incorporation in any future updated versions of the inventory.

**Response to Short Comment 2 (**Gabriel Oreggioni)

We would like to thank Gabriel Oreggioni for the comments. It is our understanding that the goal of the comments is to provide additional information to the reader rather than a critique.

In regards to the new version of EDGAR, we are thankful for the information and now cite EDGAR v5.0 as the most recent version available. We compare to EDGAR v4.3.2 in our work because it was the version available during inventory construction but it will be useful to compare the fuel specific emissions available with EDGAR v5.0 with our work in the future.

In regards to the difficulty of oil/gas estimates, we whole-heartedly agree that there is significant uncertainty in estimates of oil/gas emissions and recognize that the focus of EDGAR is on consistent methodology across temporal, spatial, and sectoral scales for multiple atmospheric species. We appreciate the information and assistance that the EDGAR team has provided regarding the EDGAR methane emissions product and will continue to contact the EDGAR team with any questions we

15 have.

[revised manuscript text omitted]